# A CitSci Approach for Rapid Earthquake Intensity Mapping: A Case Study from Istanbul (Turkey)

**Ilyas Yalcin [1,2], Sultan Kocaman [1] and Candan Gokceoglu [3,*]**

[1]   Department of Geomatics Engineering, Hacettepe University, Beytepe Ankara 06800, Turkey;
      ilyas.yalcin@hacettepe.edu.tr (I.Y.); sultankocaman@hacettepe.edu.tr (S.K.)
[2]   Gemlik Asim Kocabiyik Vocational School, Bursa Uludag University, Gemlik Bursa 16600, Turkey
[3]   Department of Geological Engineering, Hacettepe University, Beytepe Ankara 06800, Turkey
[*]   Correspondence: cgokce@hacettepe.edu.tr

**Abstract:** Nowadays several scientific disciplines utilize Citizen Science (CitSci) as a research approach. Natural hazard research and disaster management also benefit from CitSci since people can provide geodata and the relevant attributes using their mobile devices easily and rapidly during or after an event. An earthquake, depending on its intensity, is among the highly destructive natural hazards. Coordination efforts after a severe earthquake event are vital to minimize its harmful effects and timely in-situ data are crucial for this purpose. The aim of this study is to perform a CitSci pilot study to demonstrate the usability of data obtained by volunteers (citizens) for creating earthquake iso-intensity maps in a short time. The data were collected after a 5.8 Mw Istanbul earthquake which occurred on 26 September 2019. Through the mobile app "I felt the quake", citizen observations regarding the earthquake intensity were collected from various locations. The intensity values in the app represent a revised form of the Mercalli intensity scale. The iso-intensity map was generated using a spatial kriging algorithm and compared with the one produced by The Disaster and Emergency Management Presidency (AFAD), Turkey, empirically. The results show that collecting the intensity information via trained users is a plausible method for producing such maps.

**Keywords:** citizen science; CitSci; earthquake; intensity mapping; disaster mitigation; spatial kriging

---

## 1. Introduction

Sustainable development is the major aim for all countries, and the advancements in geospatial and mobile technologies provide the essential support to improve humans' lives and to protect the planet. According to the "We Are Social and Hootsuite's Global Digital 2019 Report", there are 5.1 billion mobile users worldwide [1]. Moreover, the number of users has a growth rate of 2% compared to the previous year, while the majority of mobile users were encountered in Asia. Considering that most of the devices have an Internet connection, the masses can contribute to science for solving global problems using their mobile phones if they are guided correctly. Citizen Science (CitSci) is a research approach that aims to contribute to scientific processes with the help of ordinary people (i.e., non-professional scientists) and can be utilized by many disciplines. CitSci can also be defined as a type of science developed and adopted by citizens that can help in case of dangerous situations or needs [2].

The increase in the number of natural hazards can also be tied to the population rise, climate change and increased disaster awareness (i.e., event recording and monitoring). Improved disaster management and mitigation can be obtained by better coordination during or after disaster events, which helps to reduce the losses of lives and economic losses as well. The adverse effects of disasters

can be monitored after the events so that the necessary measures can be taken. A recent review by [3] demonstrates the potential of CitSci for efficient disaster management.

The disaster-related information can be collected by citizens in the form of volunteered geographic information (VGI) or CitSci [4]. Although the data collection methods can be various, social media platforms especially have often been used for this purpose. In addition, specialized CitSci apps or repositories have been coming into view increasingly. Among the social media platforms, Twitter from Twitter Inc., San Francisco, CA USA, is the most frequently appearing in the literature. In 2017, the tweets sent by the citizens after the earthquake on Lesvos Island in Greece were classified and macroseismic intensity maps of the region were created [5]. In another study, Twitter and a mobile app were used to collect high resolution data after urban flooding [6]. A convolutional neural network (CNN) algorithm was developed in order to categorize data related to flood collected from citizens. In this context, it is concluded that CitSci can be regarded as a data source used in different disciplines.

Earthquake-related studies require extensive field data both for research and disaster mitigation purposes. An earthquake may have destructive effects based on the magnitude and focal depth. It may cause damages in the infrastructure, such as building collapses, road destruction, etc., and ground deformations such as landslides, rockfalls, lateral spreads, liquefaction, surface rupture, etc. Such deformations and failures may result in more damage and more loss of lives than ground shaking. In order to understand these phenomena, which are naturally complex, timely and dense, spatial data are required. Although both CitSci and VGI methods have been employed for this purpose, CitSci can contribute to the studies at advanced levels, such as basic or even high-level interpretation by training the motivated citizen scientists, and not only the basic data collection or validation. However, preparation of training materials and giving training are immensely required for this purpose [4].

The aim of this study is to demonstrate the usability of CitSci approach for producing earthquake intensity maps with a pilot study from Istanbul Earthquake (Mw = 5.8) occurred on 26 September 2019 at 13:59 UTC. The Anatolian Plate is surrounded by North Anatolian Fault Zone, Eastern Anatolian Fault Zone, and Aegean horst-graben system and all these are seismically active. During the last two decades, several large earthquakes such as 1999 Golcuk (Mw = 7.5), 1999 Duzce (Mw = 7.2), 2003 Bingol (Mw = 6.4), 2011 Van (Mw = 7.1), 2020 Manisa (Akhisar) (Mw = 5.4), 2020 Elazig (Mw = 6.8) occurred. Following the Istanbul Mw 5.8 earthquake that is the subject of this study, approximately 150 aftershocks occurred with intensities ranging from 1.0 Mw to 4.1 Mw [7]. Although the proximity to the faults plays an important role in the intensity level felt after an earthquake, it is not the only factor. The ground conditions of the earthquake-affected area, the construction quality and the number floors of the building have strong influence on the intensity level. There is almost no analytical solution for determining the earthquake intensity in different regions and field observations are compulsory for producing reliable intensity maps.

In this study, a mobile app (named "I felt the quake") was developed for this purpose and integrated in a spatial database management system for data storage and management (Supplementary Materials). The citizen scientists participated to the study were informed about the aims of the study, the importance of their contributions and the correct use of the app. Using the collected data and spatial analysis methods (i.e., ordinary kriging), an iso-intenstiy map was created and compared with the intensity map published by The Disaster and Emergency Management Presidency (AFAD). The results show that the developed approach is useful for producing accurate and reliable iso-intensity maps, which is an important base for disaster mitigation efforts.

## 2. Background

VGI is a commonly used term in geography, which emphasizes that humans can contribute to gather geographic information thanks to their senses and superior interpretation intelligence [8]. Many studies have been conducted in order to get support from people in different scientific studies.

Brovelli et al. [9] have developed potholes and architectural barriers application related to urban monitoring and mapping of tourism points of interest, and investigated the participation of the public in

the applications of Geographical Information System (GIS). These applications were presented to users through the Open Data Kit Collect (ODK Collect) application [10], which is an Android application created with open source codes to collect data from the user through survey forms. The data were collected not only from the Android platform but also from the web with the Enketo [11] application which can work with ODK. With potholes application in the study, the broken roads in the Northern Italy region were detected and, thus, the maintenance of the potholes was accelerated. In another application on architectural barriers, it was aimed to determine the physical structures that would create undesired situations for disabled citizens in the city. In the last application developed by the same research group, points of interest for tourism were provided. Considering all of studies carried out by Brovelli et al. [9], it was clearly seen that although potholes application had more campaign time, less users showed interest in this application than other applications. It was concluded that more advertising, mapping party and gamification methods should be employed in order to spread the use of applications. In addition, it was emphasized that the familiarization of the people with the technology is of high importance for the spread of the applications. From this point of view, it was aimed to make these applications become widespread by the use of paid students under the age of thirty in the application of architectural barriers and the use of the mapping party in the tourism application.

Boyd et al. [12] conducted a study integrated with remote sensing and volunteered user data, stating that recent, spatial and reliable data were needed to end slavery activities. The study area is "Brick Belt" region which comprises North India, Nepal, Pakistan and Bangladesh. The purpose of choosing this region was the abundance of brick kilns. It was known that most of the workforce of the kilns was made up of socially excluded people. Therefore, the aim of the study was to determine the number of the kilns. High resolution satellite images of the region were obtained and the kilns on these images were marked by volunteers. In order to collect data by volunteers, a CitSci platform called Zooniverse was used. This platform has 1.6 million users worldwide as of 2020, and facilitates the collection of data required for various scientific studies. The platform serves as a bridge that connects volunteers and researchers [13]. In the study of Boyd et al. [12], the users were directed to a website called "Slavery from Space" and they made it possible to detect the kilns through images. The study was carried out with 120 volunteered users. Approximately 55,383 kilns were detected. The Rajasthan region was selected for the control study. In addition, gas emissions from these kilns and their effects on the ecosystem were discussed.

Koskinen et al. [14] conducted a study based on a participatory GIS (PGIS) approach using open source software to draw mapping of forest plantations at the regional level in Tanzania. In order to collect data, Collect Earth [15] software of Open Foris platform [16] was used. Collect Earth is an open source software developed by the Food and Agriculture Organization of the United Nations (FAO) to monitor land use and change over high spatial and temporal resolution images [17]. It helps data collection on land use and allows the worldwide use of the application since it is open source. In this study, 22 participants were requested to complete survey data entries about forest types in the study area with the help of Collect Earth software and the reference data was created. The study was concluded by interpreting the effect of user-oriented data on the classification.

Hicks et al. [18] analyzed 106 CitSci projects within the scope of disaster risk reduction and aimed to look at citizen science from a wider window and to establish principles for the correct development of this science. In this study, six principles were introduced to guide the implementation of CitSci studies in order to reduce disaster risks. Similarly, 10 principles were introduced by the European Citizen Science Association (ECSA) [19] and translated into 30 languages. In the study, a new definition for citizen science was provided by replacing the word "information" with the word "science", since it was thought that the data collected by the citizen scientists may be in the social and cultural context and they may contribute to the applications on the focus subject. In addition, an online map which provides country based demonstrations of citizen science projects was created within the website developed in the study [20]. The website can be defined as a collective and extensive in which anybody

can follow the all studies focusing on reducing disaster risks, necessary briefing can be provided, and studies can be made widespread.

Liang et al. [21] aimed to increase their contribution to earthquake research by raising the awareness of Taiwanese people on earthquakes through the applications they developed. In this way, with the awareness of the public, frequent earthquakes in Taiwan can be survived with least damage. The Earthquake Science Information (TESIS) [22] web application developed by the group provides earthquake reports from Taiwan official institutions. This platform both helps the public to have the credible information after an earthquake and the scientists to identify the fault lines and other hazards that could lead to other hazards. They developed two applications to enable CitSci. The first one shows earthquake intensity by determining the weight from the data obtained from surveys. This application was designed based on "Did You Feel It" (DYFI) [23] developed by the United States Geological Survey (USGS). The DYFI application is a data collection platform that allows people to provide earthquake-related information through online surveys, which can then be used to create earthquake intensity maps. In this application, it was seen that there were differences between user data and institutional data. Anticipating that the differences result from the length of the survey given to users, they decided to shorten it. Secondly, sensors were installed on the computers of more than 200 educators in Taiwan, and data on earthquake waves were collected online. With the publication of the collected data as an educational argument, it may be guiding scientific studies. In addition, unlike these studies, users were trained and an Ushahidi-based application was developed to enter the situations that may occur after the earthquake such as cracks. Ushahidi [24] is a paid online application that helps collect data like ODK. This application can facilitate the management and reporting of data from mobile platforms. In this study, more than a hundred users were trained and a sample reporting application was implemented.

Kong et al. [25] developed a mobile app called MyShake [26] which could warn about earthquake shocks in advance. In the app, the accelerometer signals on mobile devices were analyzed by an artificial neural network (ANN) algorithm to determine if they point out an earthquake event or not. The signals were recorded when mobile devices were at a fixed position (e.g., hands-free on a table) and compared with the signals at the time of an earthquake. The ANN method utilized the changes in the amplitude and the frequency of the signals. In the study, it was emphasized that citizens can perform this task with their mobile devices, which can be particularly useful in countries without an earthquake early warning system.

A mobile and web-based CitSci application, called LaMA, was developed to detect landslide incidents [27,28]. In the application, photographs and the location of a landslide observed by the users were collected in a database. The data can be collected by users via web browser and a CNN architecture was developed for verification of landslide photographs as well [29]. In order to train the CNN algorithm, images were obtained from various data repositories and websites and it is possible to classify the landslide photographs with high accuracy.

AFAD of Turkey developed an earthquake mobile application called eAfad [30]. The purpose of the application is to inform citizens about earthquakes in Turkey and the surrounding areas [31]. The data are being taken from the Earthquake Data Center. Information on earthquakes with a magnitude of 4.0 Mw or larger are disseminated by the app by AFAD. The app also has features for visually impaired people. The first results of the application were presented by Eravci et al. [32]. On the other hand, a Web-GIS framework based on open source technologies was proposed by [33,34] with a mobile app for data collection with the theme of earthquake data collection, but neither practical implementation for open use nor a case study existed.

## 3. Data Collection and Analysis Methodology

### 3.1. System Design and Implementation

The mobile earthquake app ("I felt the quake") developed in this study works on an Android platform and has been used to collect data regarding the location, date, time, the intensity level as well as the name of the data provider who felt the quake (Table 1). The name field is, however, filled out optionally. The data have been recorded in a spatial database. The date and time fields in the app represent the event occurrence time, whereas the data upload (transaction) time is also sent to the database automatically. The geographical location, where the person was present at the time of earthquake, must also be entered by the user by selecting on a map (Figure 1). After pressing the save button, the name, surname, date, time, user-interpreted intensity value, transaction date and time (taken from the mobile phone system), and the location (geographical coordinate values) are sent to the server and constitute a tuple in the spatial database.

**Table 1.** Data fields and types collected in the developed app.

| Column | Data Type |
|---|---|
| Id | Integer |
| Name | Character |
| Surname | Character |
| Intensity | Character |
| Latitude | Double precision |
| Longitude | Double precision |
| Date | Character |
| Time | Character |
| Transaction date and time (*not shown on the interface*) | Timestamp |

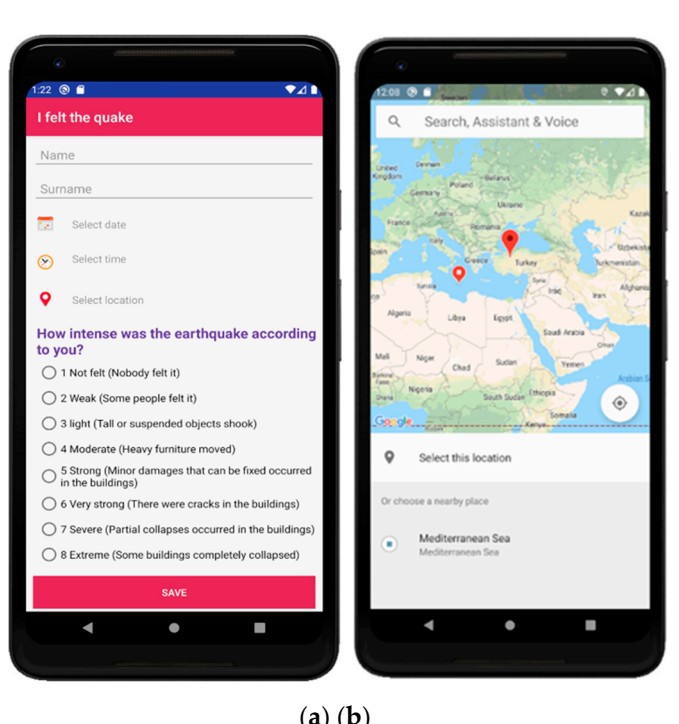

(**a**) (**b**)

**Figure 1.** (**a**) Interface of the 'I felt the quake' application; (**b**) a screenshot from 'Select location' interface.

The intensity values shown in Figure 1a and explained further in Table 2 are revised from the Modified Mercalli Intensity Scale (MM) developed in 1931 by Wood and Neumann [35]. The MM scale is also in use by USGS [36]. In order to design the interface effectively and use only the meaningful

values for the participants, the scale values were reduced to 8 as shown in Table 2. In the MM, the values 1 and 2 are too weak to be felt by a person, and the values 11 and 12 are too strong and destructive, so that the use of the app would be unnecessary. Therefore, these four values were not included in the app to simplify the interface.

**Table 2.** The relationship between Modified Mercalli Intensity Scale and values used in the developed app.

| Scale Value in the App | Description of the Intensity Scale in the Developed App | Corresponding Modified Mercalli Intensity Scale |
|---|---|---|
| 1 Not felt | Nobody felt it | 3 |
| 2 Weak | Some people felt it | 4 |
| 3 Light | Tall or suspended objects shook | 5 |
| 4 Moderate | Heavy furniture moved | 6 |
| 5 Strong | Minor damages that can be fixed occurred in the buildings | 7 |
| 6 Very strong | There were cracks in the buildings | 8 |
| 7 Severe | Partial collapses occurred in the buildings | 9 |
| 8 Extreme | Some buildings completely collapsed | 10 |

As a spatial database management system (DBMS), the open source PostgreSQL with PostGIS [37] extension, which enables spatial functionality in the DBMS, is utilized here. The selection was based on open source environment and enabling spatial queries via indices. The app was developed in an Android Studio IDE (Integrated Development Environment). It can be installed on mobile devices that support Android 4.0.3 and further versions from the Google Play Store website [38]. While designing the application interface, it was aimed to keep it simple and comprehensible for non-expert users. Further attention was paid to keep the application size (storage requirement) relatively small. The geolocation information entry by the citizen scientists was enabled using the Google-powered PlacePicker API (Application Programming Interface) in order to fully provide the application with backward data entry while not weakening the interface simplicity. Since the case study focuses on Istanbul and surrounding provinces, the app was prepared in Turkish as default with English language support. The overall system architecture design is presented in Figure 2. In the system architecture in Figure 2, the working direction of the system is defined by arrows. Information about the earthquake intensity felt by the user is entered manually to the app. The connection between the Android app and the server is provided via the Internet. A web service is used to provide the connection between the Android app and PostgreSQL database. Amazon Web Services (AWS) from Amazon.com Inc., Seattle, U.S.A. was used as the server system to ensure continuous operation of the system. Input data provided from the user is recorded in the database management system installed on the server. Then, the toast message is returned to the user indicating that the provided data has been saved.

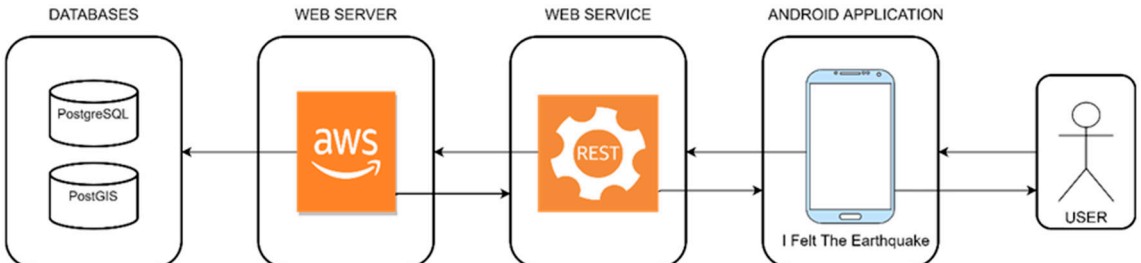

**Figure 2.** System architecture design of the application.

The application was activated on Google Play Store on 12/10/2019. All participants were informed about the main goals of the study and instructed on the use of the app personally, since external validation of such data is almost impossible due to the nature of the problem (i.e., different ground conditions and being on different floors may cause variations in the intensity level felt by the user).

Although more data came from various regions across Turkey that are not only from the trained users, the data was filtered for the trained users via name and surname. On the other hand, only the data of the Marmara Region were employed in the analysis and the results. Similarly, the Taiwan Scientific Earthquake Reporting System (TSER) provides training to their users in order to ensure the data quality [21]. The difficulties in determining the volunteers' profiles (e.g., background, knowledge and skills, etc.) or ensuring the soundness of the data were reported as known issues in CitSci studies as stated by [39].

*3.2. Geostatistical Analysis for Iso-Intensity Map Production*

Geostatistical analysis was applied to the earthquake data collected with the app in order to produce a continuous iso-intensity map. Geostatistical approaches are often used to identify and analyze spatial changes in natural phenomena, and enable statistical determination of the spatial relationships among sample data in a region. Such approaches cover the mathematical and statistical principles generally used by experts in geology and mining [40]. In the field of geostatistics, Krige's work in the field of mining in the Witwatersrand region of South Africa is accepted as pioneering [41]. Matheron emphasized the importance of regional variable for geostatistics [42].

Kriging, one of the spatial interpolation methods, is widely used in geostatistical analysis studies. There are many types of kriging interpolation methods, such as ordinary kriging, simple kriging, and universal kriging [43]. Kriging interpolation is a method that estimates the values of unknown data by statistically employing the values of sample data in an area. This method uses semivariance values between point pairs. Semivariance reflects the degree of uniqueness between point pairs with increasing distance from covariance [40]. The graphic on which semivariance represented is called variogram. Three active terms define a variogram, which are nugget (C0), range (C1), and sill (C0 + C1). The nugget is used to identify the discontinuity encountered in the inability to detect similarity between points close to each other. Range is used to define the distance required to reach the variogram threshold. Beyond this distance, location-based dependence ends. Sill is the maximum value that the variogram reaches. Variogram will take values around the sill value after it reaches to the sill value [44].

Using the statistical calculations in Kriging method, the variance for each unknown point is calculated, which indicates the reliability of the interpolation. It also enables weight calculations for unknown points from known points through semi-variogram. In order to obtain neutral results, it is restricted to Lagrangian multiplier ($\lambda$). Therefore, the sum of the weights obtained is expected to be equal to one [44]. From this point of view, by looking at the weight of each unknown point, it is possible to establish a distance–proximity relationship to the known points. In other words, while the highly weighted points are close, the low-weighted points remain distant.

In this study, location-based earthquake data were modeled using the ordinary kriging interpolation method. Ordinary kriging has similar aspects to simple kriging. In the ordinary kriging method, the local mean is not known within the search area, but taken as constant [40].

## 4. Results Obtained

This study was based on the data collected between 12/10/2019–13/11/2019 after the Istanbul earthquake (Mw = 5.8). During this period, a total of 156 records were obtained from the app. Out of those, 99 of them were provided by the trained users. The iso-intensity map of the study was produced using the spatial kriging method and compared with the one produced by AFAD Earthquake Department responsible for the Marmara Region [7]. The iso-intensity map produced by AFAD is shown in Figure 3. The map was georeferenced in the study according to The European Petroleum Survey Group (EPSG) 4326 projection in order to match with the projection system of the location data collected from the users using a number of 2D ground control points extracted from existing maps. The data points collected in the study are denoted on the AFAD map in Figure 4. It should be noted that all 12 MM intensity values exist on the legend of the map produced by AFAD. The differences

between user data and intensity map produced by AFAD can be seen clearly, as expected due to the nature of the problem.

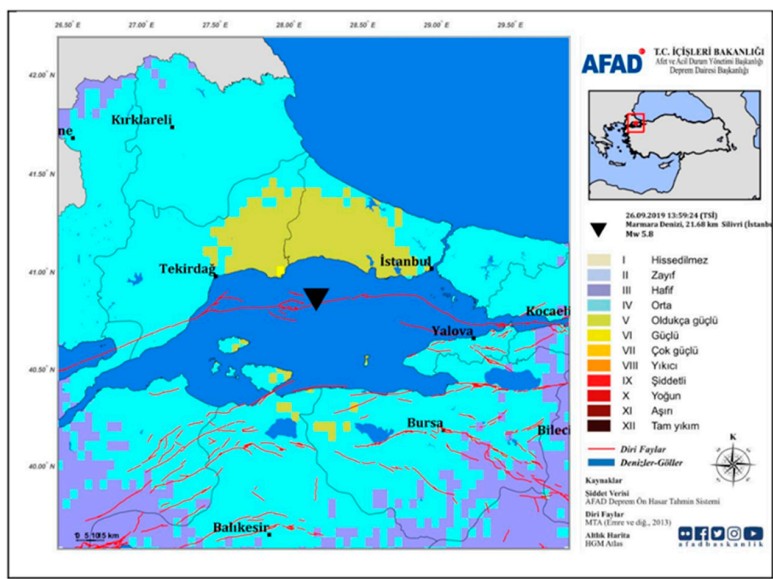

**Figure 3.** AFAD Earthquake Intensity Map produced after the Istanbul Earthquake (Mw = 5.8) [7].

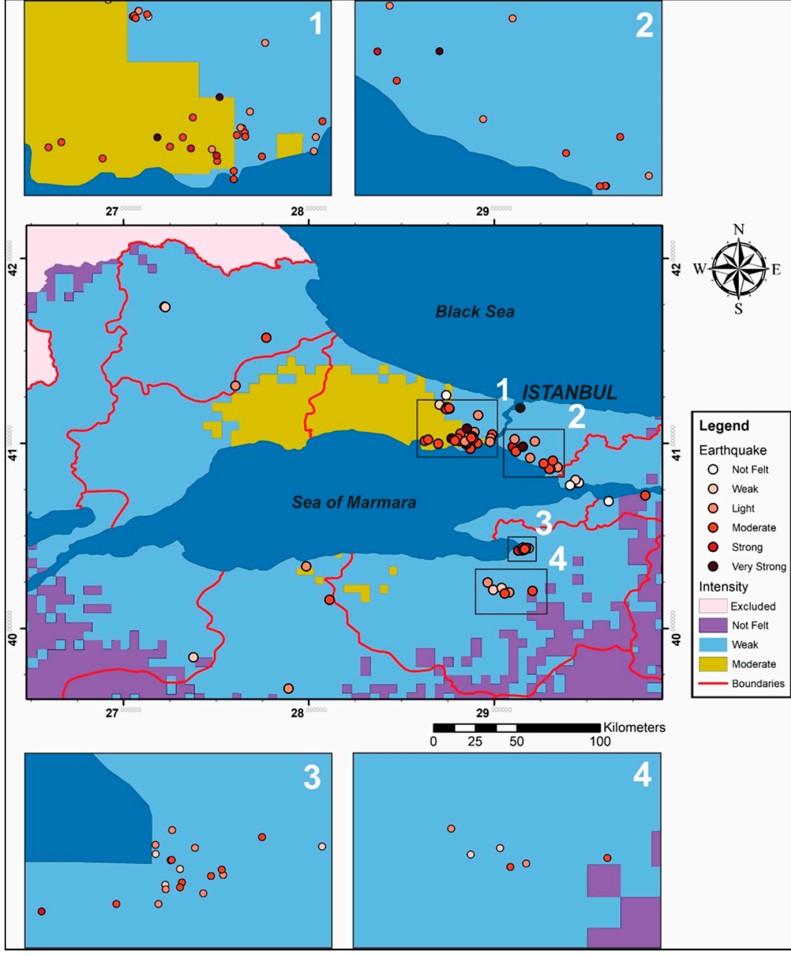

**Figure 4.** Data points collected in the study and AFAD Iso-intensity map; additional zoomed views for four sub-parts.

A statistical analysis between of the study data provided by the 99 volunteers and the corresponding value on AFAD intensity map is given in Table 3. As can be seen in the table, the scale 2—Weak (Some people felt it) largely complies with the AFAD map (15 out of 16 data points are in accordance with the map). Many points with scales of 3—Light and 4—Moderate are located on the Weak areas according to the AFAD map, as shown in Figure 4. A total of seven data points came with Strong and Very strong scales (Table 3), although these scales did not exist in the AFAD map. However, the volunteers who provided these values stated that these buildings were marked as severely damaged by the relevant municipalities and the decisions for demolition were taken. Examples photos from these buildings were also provided by the volunteers as proof. In Figure 5, two sample photos from one building which provided the scale value as Strong are shown. In Figure 6, a photo from another building with Very Strong scale is provided.

The resulting iso-intensity map obtained after spatial interpolation with ordinary kriging method can be seen in Figure 7. The minimum and the maximum intensity values obtained from the kriging method in the map area are 4.7 and 6.6, respectively.

**Table 3.** The number of intensity values obtained on the AFAD map and the data collected in the study.

| Frequency | App Scale Number | Number of Compatible Scale Values (cells) in AFAD Intensity Map |
|-----------|------------------|----------------------------------------------------------------|
| 6 | 1 (Not felt) | 0 |
| 16 | 2 (Weak) | 15 |
| 29 | 3 (Light) | 1 |
| 41 | 4 (Moderate) | 7 |
| 4 | 5 (Strong) | No data |
| 3 | 6 (Very strong) | No data |

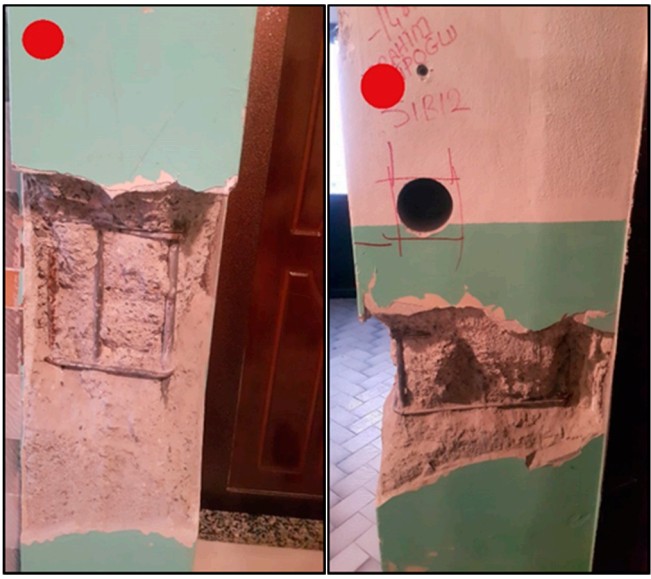

**Figure 5.** Column damages of a building to be demolished. Photos were provided by the study participant upon request of the authors. The intensity value provided by the participant was "Strong".

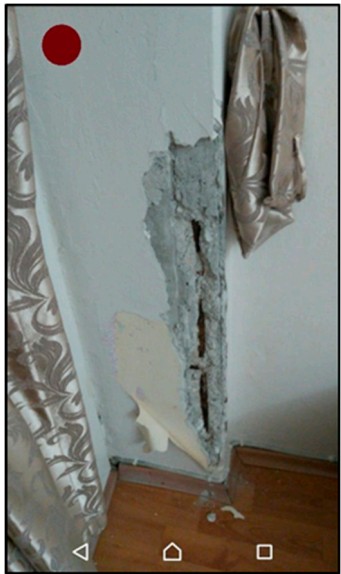

**Figure 6.** Image sent by a user after the earthquake. The intensity value provided by the participant was "Very Strong".

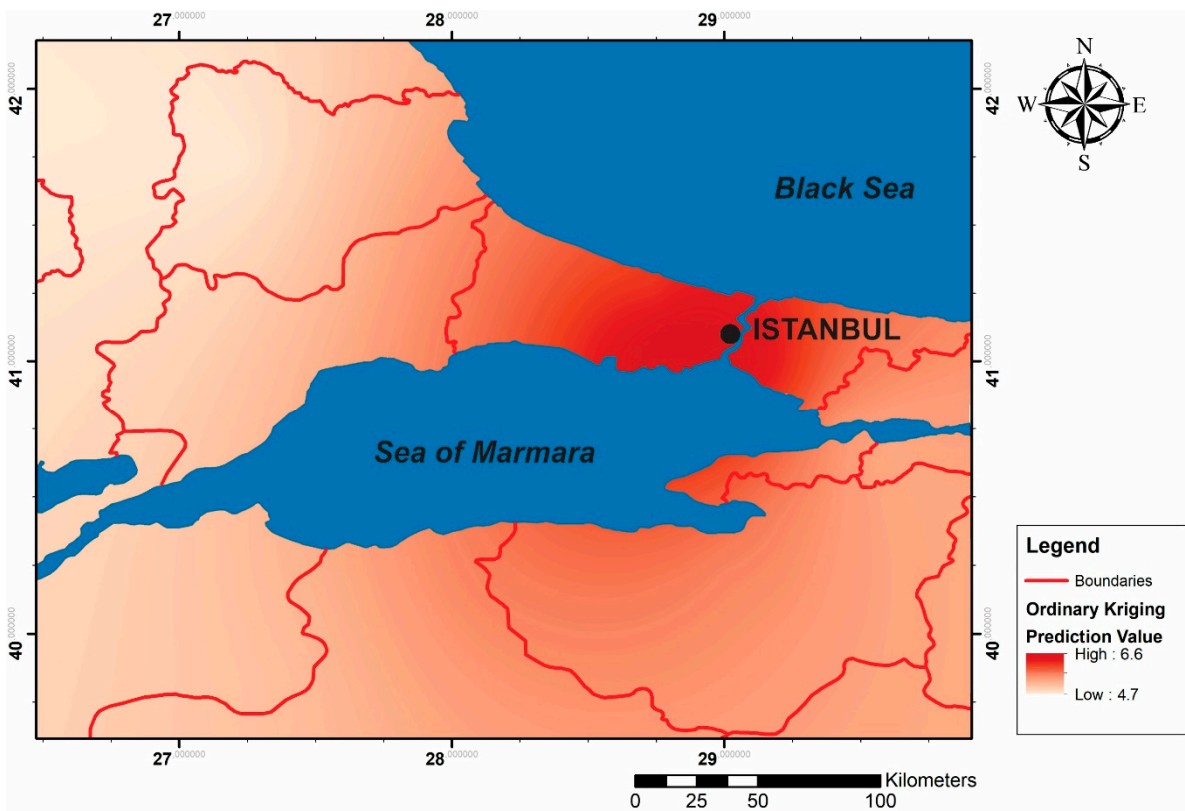

**Figure 7.** Interpolation map produced from the data collected in the study using the ordinary kriging method.

Figure 8 shows the drawing of the variogram obtained from the application data. The horizontal axis represents the distance (in degrees) that each point pair, which is used to calculate the intensity variance, has. The variances in the user-provided intensity values range between 0.1–2.4 (i.e., between 0.05–1.2 for semi-variance). As expected, there are high similarities between the intensity values at close distances. On the other hand, both small and high variations are observed at large distances due

to the radial behavior of the intensity values. The intensity values point pairs located at far distances from the earthquake epicenter are similar and small. On the other hand, the point pairs having a point close to the epicenter and the other point very far exhibit larger variations.

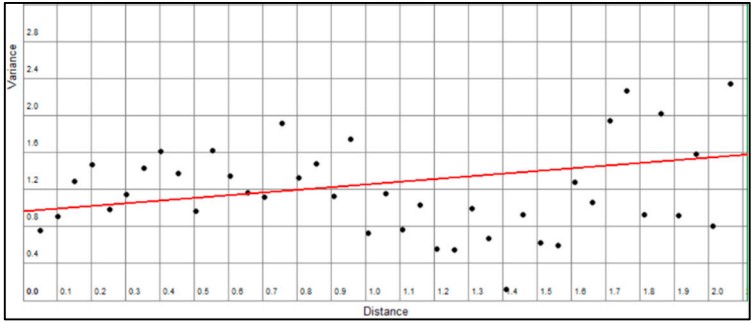

**Figure 8.** Variogram produced from the data collected in the study.

## 5. Discussion

### 5.1. Production of Intensity Maps

In this study, differences were observed between the map produced by AFAD and the information received from the citizen scientists. Production of reliable earthquake iso-intensity maps is crucial for disaster mitigation efforts and the increasing use of CitSci methods and platforms will support the development in this field. Therefore, citizens need to be informed about the importance of their high-quality contributions.

Data quality is an important aspect of CitSci studies. The term quality in this study involves correctness of the location data and the other information content, and the spatial distribution of the collected data. Inadequacy in the number and distribution of the data points would lead to weak conclusions, although the proposed kriging method can handle this issue to some extent. On the other hand, provision of training to the volunteers regarding the purpose of the project, the consequent stages and the outcomes is crucial for the success of the study.

### 5.2. The Use of the CitSci Method and the Issuess (Lessons Learned)

The widely used mobile devices can bring people who have similar experiences after a disaster event together to share their ideas. Surveys show that there is an increasing dialogue between people having similar ideas with the use of mobile technologies. Even in underdeveloped societies, the use of mobile technologies may facilitate scientific research with CitSci, and people can be connected at advanced levels in comparison to sensors. Instead of using costly sensors, each human can act as a sensor to bring a new dimension to scientific studies. In addition, they can help to reduce the costs depending on the scale of the studies. Furthermore, although people can collect data on a specific subject just like a sensor, they can also produce information on diverse subjects. Thus, instead of establishing different infrastructure for multiple studies, the required information can be obtained from the CitSci network, which can be the case for earthquakes that trigger further events (e.g., landslides).

Increasing the number of CitSci studies can help to advance the technology and the public awareness of disasters. By training the volunteers in the scope of different CitSci studies and producing data in these studies can provide better insight on different problems of the society and scientific solutions. Thus, not only the information society that produces data, but also the elevated information society, which has a high perception and understanding on many issues, can be created. Awareness of people with the increase of knowledge can protect the world against many issues, especially the environmental ones. In addition, when conducting CitSci studies, it may not be required to classify people according to their level of education, because the trainings given before the studies and the

data obtained can be passed through some filters to produce information from people at all levels. This allows people from various educational backgrounds to participate in CitSci projects.

When the proposed approach, i.e., using specially designed applications for data collection, is compared with social media-based disaster data collection efforts, the main advantages are collection of structured data with correct geolocation information via trained users that allows rapid processing and design flexibility. A study on using crowd sourcing information collected from Twitter has shown that several issues were encountered with the existence of geotags, spams, information volume, linguistic differences, and insufficient geographical coverage [45]. Another study on using Twitter data for flood assessment has also shown that geographical location of tweets can be different from the event location, the geographical distribution can be an issue, and finding the relevant information inside a large numbers of tweets is challenging as well [46]. On the other hand, the disadvantages of using a specifically developed app could be listed as the maintenance of the app, the need of multi-platform (e.g., operating system) support, and convincing the users to install and use an additional app on their mobile phones.

## 6. Conclusions

In this study, the iso-intensity map of the Istanbul 5.8-Mw earthquake, which occurred on 26 September 2019, was produced based on CitSci data and the ordinary kriging method for spatial interpolation. A total of 99 data points collected over the Marmara Region of Turkey were employed for this purpose. Prior to the data collection, the volunteers were informed about the main goals, the research problem and the expected outcomes of the study. The intensity scale used here was a revised form of Mercalli scale and the data collection was performed using an Android mobile app developed in the study. The data upload was performed online to the server which has a spatial DBMS installed.

The results show that the proposed intensity scale is suitable for producing iso-intensity maps rapidly because the iso-intensity maps are extremely important for disaster management efforts after a large earthquake. However, the assessment of the intensity of a quake at a given location used to be a slow process, as it was usually performed by means of personalized surveys [47]. For this reason, some researchers (i.e., [47–50]) developed some empirical approaches to assess intensity based on some earthquake parameters such as peak ground acceleration (PGA), peak ground velocity (PGV), peak ground displacement (PGD), the magnitude scale, and the epicentral distance, etc. However, the methodology presented here is different and based on volunteer observations. The methodology is quite reliable considering the facts that the citizen scientists were informed about the importance of their high-quality contributions. Due to the nature of the problem, it can be said that CitSci is the only reliable source of data (apart from field work by experts) to produce the iso-intensity maps since the intensity levels depend on the local ground conditions, construction date and quality, and the number of floors where the person was located at the time of the earthquake event. The differences between the intensity map published by AFAD and the citizen collected data confirms the conclusion. Yet, an estimate of differences between areas demonstrated by the intensity map is useful.

Since the citizen scientists are the key to the data quality, provision of the necessary training and technological tools (e.g., specially designed apps) would increase the reliability of such studies utilizing CitSci methods. In addition, supplementary procedures, machine learning and data analysis methods can support the data validation in large scale CitSci projects. Further spatial and logical analysis could also be employed as automatic quality control procedures, such as using a function of the measured earthquake intensity and the geographical location of the provided data as an indicator of the possible values felt by people together. When large amounts of contributions are provided by users, it is also expected to have a normal distribution of errors and, thus, the outliers can be eliminated by the ordinary kriging method and a smooth iso-intensity map can be obtained.

It can also be said that the number of such scientific studies may increase in the future. As a result, CitSci will make more promises in the future by incorporating human efforts with the technological advancements.

**Supplementary Materials:** The mobile app "Sarsintiyi Hissettim—I felt the quake" is available online at Google Play. Available: https://play.google.com/store/apps/details?id=com.ilyas.asus.postgresqlsample2&hl=en_US.

**Author Contributions:** Conceptualization and Validation: Candan Gokceoglu; Methodology, Ilyas Yalcin and Sultan Kocaman; Software and Data Curation, Ilyas Yalcin; Formal Analysis and Supervision, Sultan Kocaman; Writing-Original Draft Preparation, Ilyas Yalcin; Writing-Review & Editing, Sultan Kocaman and Candan Gokceoglu. All authors have read and agreed to the published version of the manuscript.

**Funding:** This research received no external funding.

**Acknowledgments:** The authors gratefully acknowledge AFAD for provision of the detailed earthquake report and the volunteers who provided their honest opinions about the earthquake event.

**Conflicts of Interest:** The authors declare no conflict of interest.

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
