# Peer review of "A CitSci Approach for Rapid Earthquake Intensity Mapping: A Case Study from Istanbul (Turkey)"

_ijgi, doi:10.3390/ijgi9040266_

Round 1
Reviewer 1 Report
I have no comments for authors.
Author Response
Dear Reviewer,
We would like to express our gratitude to you for taking your valuable time to review our manuscript.
Sincerely,
Authors
Reviewer 2 Report
This study identifies the characteristics of disasters among studies based on disaster information and believes that it is closely related to decision-making including dissemination of situations and evacuation to citizens. This research paper certainly has academic value. In this regard, the most important factor is the greatest justification for how to identify and collect and provide information when an earthquake event occurs.
I will list the review opinions as follows.
1. If you look at some of the contents mentioned in this paper, in the event of an earthquake, the geographic information of the respondents and the characteristics of the disaster are selected and entered directly. Personally, I need to worry a little more about how reliable and accurate the information collected can be, and how secure this information can be in an urgent situation.
2. Recently, in the case of "crawling", which collects information such as "SNS" that is frequently used by citizens for the dissemination of disaster situations, studies have been conducted to collect and disseminate the characteristics of disasters mentioned on the web through a server. Unlike these studies, it is necessary to clearly describe the merits of this study. It may also help to clarify constraints if necessary.
3. The picture in Figure 8 needs to be explained more. If possible, it would be a good idea to elaborate on the correlation.
Author Response
Dear Reviewer,
We are thankful for taking your time to review our manuscript and giving valuable comments. We believe that the quality of our manuscript has been improved significantly after revision. Attached please see our replies to your comments.
Kind regards,
Authors

Reviewer 3 Report
This paper introduces a rapid earthquake intensity mapping method utilizing Citizen Science. The authors developed a mobile APP “I felt the quake” to collect earthquake intensity information from various locations with the aid of citizens. The iso-intensity map was then generated for disaster management. The exploration of the CitSci approach is a worthwhile research and I was excited to read about this work. However, many details are not elaborated clearly in the current version. So, I recommend that this paper be accepted with major revision.
Major issues:
- Could you please explain why only 99 volunteer data are used in the experiment? In my opinion, Citizen Science is useful since it makes various volunteer data available. If users of the App need training and data provided by the non-trained users are filtered out, how can you make readers think your App is useful?
- In the system design section, please add details to introduce the 8 intensity types. Besides, I think it is hard for users to differentiate some intensity types, even for trained users. How will you deal with this kind of problem?
- L300 In figure 4, the legend only shows 6 types of intensity, but in the previous sections, the authors said they use 8 types of intensity. Why?
Additional comments:
L238: Architecture shown in Figure 2 is not introduced in the paper.
L328: How did the authors know the minimal and maximal intensity value?
Author Response
Dear Reviewer,
We are thankful for taking your time to review our manuscript and giving valuable comments. We believe that the quality of our manuscript has been improved significantly after revision. Attached please see our replies to your comments.
We hope that the revisions may satisfy you.
Kind regards,
Authors

Round 2
Reviewer 2 Report
It is judged that the publishing stage has been approached through the review process.
Reviewer 3 Report
The authors addressed my comments.